# Clinical Relevance of Myopenia and Myosteatosis in Colorectal Cancer

**DOI:** 10.3390/jcm11092617

**Published:** 2022-05-06

**Authors:** Yoshinaga Okugawa, Takahito Kitajima, Akira Yamamoto, Tadanobu Shimura, Mikio Kawamura, Takumi Fujiwara, Ikuyo Mochiki, Yoshiki Okita, Masahiro Tsujiura, Takeshi Yokoe, Masaki Ohi, Yuji Toiyama

**Affiliations:** 1Department of Genomic Medicine, Mie University Hospital, Tsu 514-8507, Japan; north.at.island@gmail.com (T.K.); t-fujiwara@med.mie-u.ac.jp (T.F.); i-mochiki@doc.medic.mie-u.ac.jp (I.M.); 2Department of Gastrointestinal and Pediatric Surgery, Division of Reparative Medicine, Institute of Life Sciences, Mie University Graduate School of Medicine, Tsu 514-8507, Japan; numberbee8@gmail.com (A.Y.); tadanobu189222@gmail.com (T.S.); up2date0521@gmail.com (M.K.); nyokkin@clin.medic.mie-u.ac.jp (Y.O.); masahirotsujiura@med.mie-u.ac.jp (M.T.); takesche@clin.medic.mie-u.ac.jp (T.Y.); mohi1012@med.mie-u.ac.jp (M.O.)

**Keywords:** sarcopenia, survival, infectious complications, adverse effects, colorectal cancer

## Abstract

Sarcopenia was initially described as a decrease in muscle mass associated with aging and subsequently also as a consequence of underlying disease, including advanced malignancy. Accumulating evidence shows that sarcopenia has clinically significant effects in patients with malignancy, including an increased risk of adverse events associated with medical treatment, postoperative complications, and a poor survival outcome. Colorectal cancer (CRC) is one of the most common cancers worldwide, and several lines of evidence suggest that preoperative sarcopenia negatively impacts various outcomes in patients with CRC. In this review, we summarize the current evidence in this field and the clinical relevance of sarcopenia in patients with CRC from three standpoints, namely, the adverse effects of medical treatment, postoperative infectious complications, and oncological outcomes.

## 1. Introduction

Colorectal cancer (CRC) is one of the most common gastrointestinal cancers and continues to be the second leading cause of cancer-related deaths worldwide [1]. Although improved surgical techniques, including laparoscopic and robotic-assisted surgery, have emerged and progressed in the last decade, surgical morbidity and mortality remain one of the clinical hurdles to be overcome in patients with CRC undergoing surgery. Furthermore, the survival outcome remains poor, especially for CRC with metastatic disease, despite significant advances in the medical and radiological treatments available for advanced CRC. Different approaches, including nutritional intervention and rehabilitation, are urgently needed to improve the overall health status of patients with CRC, optimize their treatment, and improve their prognosis.

Sarcopenia was initially defined as an age-related loss of skeletal muscle mass and function and reported to be present in approximately 6–22% of older adults [2]. The word comes from the Greek sarx (meaning “flesh”) and penia (meaning “loss”) and was coined by Rosenberg in 1989 [3]. Sarcopenia may be primary or secondary. Primary sarcopenia is the classical form is understood to be part of the aging process and precedes the onset of frailty. In contrast, secondary sarcopenia is caused by various pathogenic mechanisms that may be disease-related (e.g., inflammatory disease and malignancy), activity-related (e.g., disuse), or nutrition-related (e.g., protein deficiency) [4,5].

## 2. Definition of Sarcopenia

Several international groups, including the Society of Sarcopenia, Cachexia, and Wasting Disorders [6], International Working Group on Sarcopenia [7], Asian Working Group for Sarcopenia [8], European Society of Parenteral and Enteral Nutrition [9], and European Working Group on Sarcopenia in Older People (EWGSOP) [10], have proposed various definitions and diagnostic criteria for sarcopenia. For example, the EWGSOP proposed a definition of sarcopenia in 2010 that aimed to foster advances in identifying and caring for people with sarcopenia and published a revised consensus definition of sarcopenia and updated diagnostic criteria in 2018. According to the updated statement, the most important criterion for a diagnosis of sarcopenia is low muscle strength, which is identified primarily by grip strength or the chair stand test (criterion 1). The diagnosis is confirmed by the finding of low muscle quantity and quality using any validated technique, such as dual x-ray absorptiometry, bioelectric impedance analysis, or computed tomography (CT) (criterion 2). Poor physical performance, as determined by an assessment of gait speed or the chair stand test, is indicative of severe sarcopenia (criterion 3). The clinical algorithm in this revised consensus was updated for case finding, diagnosis, confirmation, and determination of severity. Furthermore, to identify and characterize sarcopenia, optimal cut-off points for the measurement of variables have been proposed based on accumulating evidence. However, there is still no universal definition of sarcopenia, and almost all of the available data in patients with CRC comes from studies in which body composition status was assessed on CT images as a surrogate marker for sarcopenia. Based on the dysregulation patterning of body composition status using CT images, image-based surrogate markers of sarcopenia are currently recognized as belonging to two distinct types, namely, “myopenia” and “myosteatosis”. This review article investigates these indices in detail to clarify the clinical relevance of myopenia and myosteatosis in patients with CRC.

## 3. Pathophysiology of Sarcopenia in Patients with Malignancy

Accumulating data are revealing various mechanisms that may be involved in the onset and progression of low muscularity and sarcopenia in patients with malignancy. In addition to a lack of physical activity as a consequence of aging and chronic disease, muscle wasting may also be caused by the dysregulation of protein synthesis, protein degradation, mitochondrial abnormalities, inflammation, oxidative stress, and the degeneration of muscle tissue as a result of satellite cell activity [11]. One of the major causes of cancer-related sarcopenia is the degradation of normal tissue function in response to systemic inflammation, which occurs as a result of host-tumor interaction and has been termed the seventh hallmark of cancer [12]. Several studies have demonstrated that tumor-derived inflammatory cytokines, including tumor necrosis factor-alpha, interleukin (IL)-1-beta, and IL-6, are involved in the pathogenesis of sarcopenia in patients with malignancy [13,14,15]. Inter-organ communication by these tumor-derived inflammatory cytokines exacerbates metabolic dysfunction in muscle and fat, thereby reducing muscularity and causing sarcopenia [16,17]. Furthermore, IL-1 may stimulate adrenocorticotropic hormone and cortisol via the activation of hypothalamic neurons and corticotrophin-releasing hormone; this link between IL-1 and the hypothalamic-pituitary axis mediates the dysregulation of catabolic effects and promotes muscle wasting and loss of appetite in patients with malignancy [14].

Recent research has highlighted the pivotal role of growth differentiation factor 15 (GDF15), a member of the transforming growth factor-beta superfamily, in patients with sarcopenia. Several studies have demonstrated significantly increased circulating levels of GDF15 in patients with sarcopenia and malignancy, including CRC [18,19,20,21]. Mechanistically, GDF15 could induce weight loss via the GDNF family receptor alpha-like (GFRAL)-Ret proto-oncogene (RET) signaling complex in neurons in the brainstem [22,23,24,25], and the antibody-mediated inhibition of GDF15–GFRAL activity has been shown to reverse excessive lipid oxidation and prevent sarcopenia in cancer-bearing mice [26].

Furthermore, there is emerging evidence indicating that circulating microRNAs derived from tumors have a pivotal role in the loss of skeletal muscle mass via the downregulation of antiapoptotic genes in skeletal muscle cells in patients with CRC [27,28].

## 4. Prevalence of Sarcopenia in Patients with Malignancy

As mentioned earlier, sarcopenia was originally described as an aging-related disease [2]. The prevalence of sarcopenia in healthy populations gradually increases with advancing age, with previous reports suggesting an increase from 9% at 45 years to up to 64% at 85 years [29]. Moreover, malignant disease is more common in the elderly, and sarcopenia may develop as a result of both disease-related and aging-related factors. Therefore, the prevalence of sarcopenia in patients with malignancy is even higher, ranging from 11% to 74% in adults [30,31,32,33,34]. In several studies that used various assessment methods, the prevalence of sarcopenia in patients with CRC ranged widely from 12% to 60% [32,35,36,37,38,39]. Broughman et al. investigated the prevalence of myopenia in 87 patients with stage I–III CRC who were aged older than 70 years by measuring the total amount of muscle on CT images and calculating the skeletal muscle index (SMI) based on patient height [32]. In that study, 60% of men and 56% of women had myopenia. Generally, the prevalence of sarcopenia is high in patients with CRC, and the clinical relevance of low muscularity in this disease has been attracting considerable research attention in the last decade.

## 5. Clinical Impact of Preoperative Low Muscularity on Perioperative Infectious Complications in Patients with CRC

Infectious complications account for much of the postoperative morbidity in patients with CRC. Surgical site infections, including superficial, deep, and organ space infections, are particularly important because of the discomfort for patients, anxiety for surgeons, costs incurred to health care systems, and their potential to delay the start of adjuvant therapy [40]. Furthermore, there is mounting evidence that postoperative infectious complications have an adverse impact on oncological outcomes in patients with CRC [41,42,43,44]. A recent meta-analysis that evaluated 43 original papers clearly demonstrated a statistically significant association between both anastomotic leak and wound/surgical site infections and unfavorable oncological outcomes in patients who had undergone surgery for CRC [44]. Therefore, there is an urgent need to identify surrogate markers that can identify patients with CRC at high risk for postoperative infectious complications.

Several studies have suggested a close association between postoperative infectious complications and preoperative malnutrition in various malignancies, including CRC [45,46,47,48]. Several lines of evidence point to a significant association between preoperative low muscularity and postoperative infectious complications after the surgical resection of CRC [35,36,47,49]. Lieffers et al. evaluated CT images for 234 patients who underwent resection of primary CRC and demonstrated that the risk of postoperative infection was greater in patients with a low SMI (myopenia) than in those with a higher SMI (23.7% vs. 12.5%; *p* = 0.025) and that the negative impact of preoperative myopenia was more pronounced in patients over 65 years of age [35]. Another group of researchers assessed skeletal muscle mass in 302 patients who were undergoing colectomy for colon cancer and demonstrated that those with denser psoas muscle tissue had fewer postoperative infectious complications (odds ratio [OR] 0.95, 95% confidence interval [CI] 0.93–0.98, *p* = 0.001) [49]. Our research group also investigated skeletal muscle mass in these patients and established a novel parameter known as “modified intramuscular adipose tissue content” (mIMAC) as a surrogate marker for myosteatosis [50]. We assessed the preoperative mIMAC on CT images for 892 patients, including 471 with CRC who underwent surgery, and found that a decreased preoperative mIMAC (decreased quantity of skeletal muscle mass) was an independent risk factor for remote infection postoperatively (OR 2.56, 95% CI 1.06–6.23, *p* = 0.038) in patients with CRC. Trejo-Avila et al. recently published a meta-analysis of 23 studies of the impact of preoperative sarcopenia on the overall postoperative complication rate and showed that patients with sarcopenia had increased risks of surgical site infection (OR 1.4, 95% CI 1.12–1.76) and postoperative mortality (OR 3.21, 95% CI 2.01–5.11) [51]. Although there has been considerable variation in the methods used to evaluate body composition status on preoperative CT images in the studies performed to date, all of the evidence indicates an intimate correlation between preoperative low muscularity and postoperative infectious complications in patients with CRC.

## 6. Clinical Impact of Pretreatment Low Muscularity on the Effects of Medical and Radiological Treatment in Patients with CRC

Several studies have shown that pretreatment malnutrition increases the risk of chemotherapy-related toxicity and adverse events during chemotherapy in patients with malignancy, including CRC [52,53,54,55,56,57]. Prado et al. prospectively evaluated patients with stage II/III colon cancer treated with 5-fluorouracil (5-FU) and leucovorin and found that low lean body mass was a strong predictor of toxicity, especially in female patients receiving 5-FU-based chemotherapy [57]. Jung et al. assessed the cross-sectional area of the psoas muscle at L4 on pretreatment CT images from 229 consecutive patients with stage III colon cancer who received adjuvant chemotherapy consisting of oxaliplatin, 5-FU, and leucovorin and identified a decrease in the psoas mass index (PMI) value to be an independent risk factor for all grade 3–4 toxicities in these patients [55]. In the setting of metastatic CRC, Sasaki et al. evaluated the skeletal muscle area (SMA) on CT images obtained before and 3 and 6 months after treatment and demonstrated that the loss of skeletal muscle mass at 3 months after treatment was significantly correlated with the incidence of adverse events and an independent predictor of a poor objective response (OR 0.2, 95% CI 0.1–0.6; *p* < 0.01) and poor progression-free survival (PFS; hazard ratio [HR] 1.6, 95% CI 1.0–2.7; *p* = 0.04).

There is also a body of research on the clinical relevance of pretreatment low muscularity in patients with rectal cancer who receive neoadjuvant chemoradiotherapy. Almost all of these studies have found that low muscularity has an adverse impact on postoperative complications and the survival outcome [58,59,60,61,62,63,64]. Olmez et al. were the first to report a close association between myopenia and the response to neoadjuvant chemoradiotherapy in patients with advanced rectal cancer [64]. They retrospectively assessed SMI on CT images obtained at the time of the initial diagnosis from 61 patients who received neoadjuvant chemoradiotherapy for locally advanced rectal cancer. Their main finding was that the pathological complete response rate was significantly higher in patients without myopenia than in those with myopenia (21.4% vs. 3.0%; *p* = 0.025). Although the clinical significance of pretreatment sarcopenia in terms of the impact on outcomes of medical and radiological treatment remains unclear, all the evidence to date suggests that pretreatment low muscularity increases the likelihood of an unfavorable outcome in patients with CRC who receive multimodality therapy.

## 7. Clinical Impact of Pretreatment Low Muscularity on Oncological Outcomes in Patients with CRC and No Distant Metastasis

Most of the available evidence indicates that preoperative low muscularity has an adverse impact on survival in patients with CRC. The first relevant population-based study was performed by Prado et al. at a cancer treatment center in northern Alberta, Canada [65]. As part of that study, which included 2115 patients with solid respiratory or gastrointestinal tumors, including CRC, the authors measured the total skeletal muscle cross-sectional area in 250 of 325 obese patients with a body mass index ≥ 30 for whom CT images were available. They showed that the functional status was poorer in obese patients with myopenia than in those without myopenia and that myopenia was an independent predictor of overall survival (OS; HR 4.2, 95% CI 2.4–7.2, *p* < 0.0001) in these patients.

A cohort study that included 220 patients with stage I–III CRC and no distant metastasis who underwent curative resection found that recurrence-free survival (RFS), OS, and cancer-specific survival (CSS) rates were significantly shorter in patients with a low SMI value on preoperative CT images than in those with a normal SMI value (5-year RFS rate, 56% vs. 79%, *p* = 0.006; 5-year OS rate, 68% vs. 85%, *p* = 0.015; 5-year CSS rate, 82% vs. 91%, *p* = 0.026) [66]. Another population-based cohort study evaluated skeletal muscle mass and radiodensity on CT images obtained at diagnosis and approximately 14 months later to clarify the impact of a dysregulated body composition pattern on survival in 1924 patients with stage I–III CRC who underwent curative resection [67]. In that study, a marked deterioration in skeletal muscle mass and muscle radiodensity was significantly correlated with poor OS (HR 2.15, 95% CI 1.59–2.92, *p* < 0.001) and CSS (HR 1.61, 95% CI 1.2–2.15, *p* = 0.002). Our group also evaluated PMI and intramuscular adipose tissue content (IMAC) on the preoperative CT images of 308 patients with CRC, including 242 with stage I–III disease who underwent curative resection, and identified a low PMI value to be an independent prognostic factor for disease-free survival (HR 3.15, 95% CI 1.8–5.51, *p* = 0.0001) in these patients. A recent meta-analysis by Sun et al. that included 12 studies in 5337 patients with non-metastatic CRC demonstrated that low muscularity predicted significant decreases in OS (HR 1.63, 95% CI 1.24–2.14, *p* < 0.01), disease-free survival (HR 1.7, 95% CI 1.24–2.31, *p* < 0.01), and CSS (HR 1.62, 95% CI 1.16–2.27, *p* < 0.01) [68].

## 8. Clinical Impact of Pretreatment Low Muscularity on Oncological Outcomes in Patients with CRC and Distant Metastasis

There is emerging evidence of a close relationship between low muscularity and disease progression in CRC. Park et al. measured skeletal muscle mass using bioelectric impedance analysis in 1270 patients at the time of first colonoscopy and found a significant association between myopenia and an increased risk of advanced colorectal neoplasia (OR 2.347, 95% CI 1.311–4.202, *p* = 0.004) [69]. Furthermore, Okugawa et al. evaluated the PMI and IMAC on preoperative CT images for 308 patients with CRC and also found a significant correlation between a low PMI value and distant metastasis (*p* = 0.007) [70]. These findings suggest the possibility of biological mechanisms linking sarcopenia and the progression of CRC. In further support of this hypothesis, there is some evidence indicating that various products secreted by metastatic tumors, including cytokines, metabolites, microRNAs, and exosomes, can influence host organs and promote myopenia in patients with malignancy [27,28,71,72,73]. Okugawa et al. also evaluated the PMI and IMAC on preoperative CT images for 167 patients with CRC and analyzed the correlation between body composition status and preoperative serum miR-21 expression [27]. They found that serum miR-21 expression was higher in patients with CRC and a low PMI than in those with a high PMI. MiR-21 is a representative oncogenic secretory miRNA [74,75], and a recent study showed that it activates toll-like receptor-7 on murine myoblasts and promotes the death of muscle cells via c-Jun N-terminal kinase activity in cancer-related cachexia [76]. Interestingly, a further study by Okugawa et al. showed that circulating miR-203 derived from distant tissues with metastasis caused myopenia by downregulating the expression of BIRC5 (survivin) in a skeletal muscle cell line [28]. Collectively, these studies point to a close association between disease progression and low muscularity via circulating miRNAs in patients with CRC.

In line with the above-mentioned reports, several studies have confirmed the negative burden of preoperative low muscularity in terms of the survival outcome in patients with metastatic CRC. Da Cunha et al. retrospectively assessed body composition status on CT images for 72 patients with stage IV CRC [77] and found that the median PFS and OS were significant shorter in those with myopenia than in those without myopenia (PFS, 7.2 months vs. 15.2 months, HR 1.78, 95% CI 1.00–3.14, *p* = 0.048; OS, 12.5 months vs. 36.7 months, HR 1.86, 95% CI 1.02–3.38, *p* = 0.043) after adjustment for the number of metastatic lesions, resection of metastasis, and performance status. Vashi et al. evaluated the SMI value at L3 on initial CT images from 112 consecutive patients with CRC [78]. In that study, the focus was on patients with stage IV CRC (*n* = 66), and the risk of mortality was found to be four times greater in patients with myopenia than in those without myopenia (HR 4.0, 95% CI 1.7–9.3, *p* = 0.001) after adjustment for nutrition status, age, and sex. van Vledder et al. evaluated the total cross-sectional area of skeletal muscle and that of intra-abdominal fat on CT images for 196 patients with CRC and hepatic metastasis [37]. They found that the median survival time was significantly shorter in patients with myopenia than in those without myopenia (23.8 months vs. 59.8 months) and that preoperative myopenia was an independent prognostic factor for OS in patients with CRC (HR 2.69, 95% CI 167–4.43, *p* < 0.001). Collectively, this evidence is consistent with that for patients with non-metastatic CRC and suggests that low muscularity occurs in patients with CRC, whether non-metastatic or metastatic.

## 9. Correlation between Low Muscularity and Systemic Inflammatory Reaction in Patients with CRC

There is now evidence of a close relationship between cancer and inflammation [79,80]. The systemic inflammatory reaction (SIR), which occurs via the tumor–host interaction, has been described as one of the hallmarks of cancer [12,81]. Various markers and combinations thereof, including cellular (whole white cell count, neutrophils, lymphocytes, platelets) and humoral (C-reactive protein [CRP], albumin) components, can be used to assess the degree of SIR, and all of the studies consistently suggest the potential of the SIR as a prognostic marker in CRC [48,82,83,84,85,86].

In contrast, cancer cachexia is a multifactorial metabolic syndrome characterized by weight loss due to sarcopenia, and over half of patients with advanced cancer have accompanying cachexia [87,88,89]. A series of in vitro and in vivo experiments have demonstrated that proinflammatory cytokines, including interleukins 1 and 6 and tumor necrosis factor, are deeply involved in the pathogenesis of sarcopenia via a mediator of the anorexia and proteolysis of skeletal muscle [90,91]. The findings of further studies have reinforced the notion of this intimate correlation, and it is now recognized that the SIR acts as a pivotal mediator of skeletal muscle depletion via the host–tumor interaction in cancer cachexia [92,93]. The Global Leadership Initiative on Malnutrition, which has reached a global consensus concerning the identification and endorsement of diagnostic criteria for malnutrition, includes SIR as one of its etiologic criteria [94,95]. Therefore, SIRs might be involved in malnutrition in various diseases, including malignancy.

Recent studies have gradually clarified the intimate correlation between SIRs and the dysregulation of the body composition status in malignancy, including CRC. Richards et al. evaluated SMI on preoperative CT images from 174 patients with operable CRC and showed a clear linear relationship between SMI and CRP values (r = −0.21, *p* = 0.005) and between SMI and albumin values (r = 0.31, *p* < 0.001) [96]. Maliezis et al. assessed the correlationship between preoperative markers of SIR and skeletal muscle parameters on CT images from 763 patients with CRC [97]. In their study, the severity of the SIR was evaluated by the preoperative neutrophil-to-lymphocyte ratio (NLR) and the albumin level. They showed that a high preoperative NLR and a low preoperative albumin level were independent risk factors for a reduced lumbar SMI value in patients with CRC (NLR, OR 1.78, 95% CI 1.29–2.45, *p* < 0.001; albumin, OR 1.8, 95% CI 1.17–2.74, *p* = 0.007). Another research group in Glasgow quantified the SMI and skeletal muscle density (SMD) in 650 patients with operable primary CRC to determine the relationship between CT-derived body composition and SIR status measured using the modified Glasgow Prognostic Scale (mGPS) score combined with CRP and albumin levels [98]. That study also found that decreased SMI and SMD values were significantly associated with an elevated mGPS score (SMI, *p* < 0.001; SMD, *p* = 0.045) and that each of these factors, namely, a low SMI and high mGPS, was an independent prognostic factor for OS in patients with CRC (low SMI, HR 1.5, 95% CI 1.04–2.18, *p* = 0.031; high mGPS, HR 1.44, 95% CI 1.15–1.79, *p* = 0.001). In a study of the correlations of immune/SIR markers with myopenia and myosteatosis in patients with CRC, Okugawa et al. evaluated the preoperative PMI and IMAC on CT images for 308 patients with CRC [99]. They also quantified preoperative levels of various markers of the SIR, including CRP, albumin, the NLR, the platelet-to-lymphocyte ratio (PLR), the neutrophil-to-platelet score (NPS), the prognostic nutrition index (PNI), and the systemic immune-inflammation index (SII), and assessed the correlation between body composition status and several markers of SIRs in these patients. Interestingly, that study showed a significant correlation between a low PMI and elevated CRP (*p* < 0.0001), SII (*p* = 0.0027), and NPS (*p* = 0.016), and a decreased LMR (*p* = 0.039), PNI (*p* = 0.0001), and albumin level (*p* = 0.0006). Multivariate analysis identified an elevated CRP level to be an independent risk factor for a low PMI (myopenia, OR 2.49, 95% CI 1.31–4.72, *p* = 0.005) in patients with CRC. Furthermore, the same study performed propensity score matching to attenuate the selection bias arising from differences in patient characteristics and confirmed a close relationship between elevated CRP and a decreased PMI. Although these studies used different markers of the SIR, their findings indicate an intimate correlation between SIR status and low muscularity in patients with CRC. Considering these findings, the SIR via a host-tumor interaction might be one of the major drivers in the pathogenesis of sarcopenia in various malignancies, including CRC.

## 10. Discussion

Accumulating evidence suggests that low muscle mass has a negative impact on surgical, chemoradiotherapeutic, and survival outcomes in patents with CRC. To reap clinically meaningful benefits from all the recent advances in treatment, a concerted effort must be made to address several key issues that contribute to low muscularity.

One of the major issues to be resolved is that the studies reported to date have used different methods to assess myopenia and myosteatosis, including different types of body composition index (e.g., SMI, PMI, or IMAC), different levels of assessment (e.g., L3 or L4), and different cut-off values, as shown in Table 1. There are still no definitive criteria for the diagnosis of low muscularity in patients with malignancy, including CRC. There is now an urgent need to establish a global consensus regarding the diagnosis of myopenia and myosteatosis using clinically relevant methods with optimal cut-off values before these indices can be used to devise a therapeutic decision-making strategy that allows personalized treatment and improves outcomes in patients with CRC.
jcm-11-02617-t001_Table 1Table 1Association between body composition status and short-term and long-term postoperative outcomes in patients with colorectal cancer.StudyTypeNumber of PatientsType of Body CompositionAssessmentLevelType of MuscularityClinical SignificanceReferencePrado, et al.CRC250TSML3MyopeniaSurvival Outcome[65]van Vledder, et al.CRC196SMICaudal End of L3MyopeniaSurvival Outcome[37]Lieffers, et al.CRC234SMIL3MyopeniaPostoperative Infectious Complication[35]Martin, et al.CRC1473SMIL3MyopeniaSurvival Outcome[100]Sabel, et al.CC302PD, SFD, TBFL4MyopeniaMyosteatosisPostoperative Infectious Complication Survival Outcome[49]Jung et al.CC229PMIL4MyopeniaChemotherapeutic Toxicity[55]Miyamoto, et al.CRC220SMIL3MyopeniaSurvival Outcome[66]Malietzis, et al.CRC805SMI, VAT, SMDL3MyopeniaMyosteatosisSurvival Outcome[101]Boer, et al.CC91TPA, TAMAL3MyopeniaPostoperative Complication[102]Black, et al.CRC339SMIL3MyopeniaSurvival Outcome[103]Choi, et al.RC188SMIL3MyopeniaSurvival Outcome[59]Sueda, et al.CRC211SMI, SMDL3MyopeniaMyosteatosisSurvival Outcome[104]Womer, et al.RC180TPASuperior Endplate of L3MyopeniaPostoperative Complication[60]Takeda, et al.RC144SMICaudal End of L3MyopeniaSurvival Outcome[61]Brown, et al.CRC1924SMA, SMDL3MyopeniaMyosteatosisSurvival Outcome[67]Okugawa, et al.CRC308PMI, IMACSuperior Aspect of L4MyopeniaMyosteatosisPostoperative Infectious Complication · Survival Outcome[70]McSorley, et al.CRC322VFI, SFI, SMI, TFIL3MyopeniaMyosteatosisSurvival Outcome[105]van Vugt, et al.CRC816SMI, SMDL3MyopeniaMyosteatosisPostoperative Complication[106]Levolger et al.RC122SMAL3MyopeniaSurvival Outcome[58]Berkel, et al.RC99TAMA3 Anatomical LevelsMyopeniaPostoperative Complication Survival Outcome[62]Dolan, et al.CRC650SMI, SMDL3MyopeniaMyosteatosisSurvival Outcome[98]Hopkins, et al.CRC968SMI, SMDL3MyopeniaMyosteatosisSurvival Outcome[107]Okugawa, et al.CRC183PMI, IMACSuperior Aspect of L4MyopeniaMyosteatosisSurvival Outcome[28]Vashi, et al.CRC112SMIL3MyopeniaSurvival Outcome[78]Da Cunha, et al.CRC72SMIL3MyopeniaSurvival Outcome[77]Dolan, et al.CRC163TPIL3MyopeniaSurvival Outcome[108]Sasaki et al.CRC249SMIL3MyopeniaChemotherapeutic Toxicity[109]Kurk, et al.CRC450SMIL3MyopeniaSurvival Outcome[110]Olmez, et al.CRC209SMI-MyopeniaDoes Not Support Prediction for Postoperative Surgical Site Infection[47]Aro, et al.CRC348SMI, SMDL3MyopeniaMyosteatosisPostoperative Complication Survival Outcome[111]Shirdel, et al.CRC974SMI, SMDL3MyopeniaMyosteatosisSurvival Outcome[112]Xie, et al.CRC132SMIL3MyopeniaSurvival Outcome[113]Wang, et al.CRC400SMIL3MyopeniaSurvival Outcome[114]Xiao, et al.CC1630SMI, SMDL3MyopeniaMyosteatosisPostoperative Complication Survival Outcome[115]Lee, et al.CRC214SMIL3MyopeniaSurvival Outcome[116]Olmez, et al.RC61SMIL3MyopeniaPathological Response for CRT[64]Chai, et al.CRC228SMIL3MyopeniaPostoperative Complication Survival Outcome[117]Kusunoki, et al.CRC471IMAC, mIMACSuperior Aspect of L4MyosteatosisPostoperative Complication Survival Outcome[50]Abe, et al.RC225PMIL3MyopeniaSurvival Outcome[63]Uehara et al.RC262PMIL3MyopeniaPostoperative Infectious Complication Survival Outcome[118]CRC: colorectal cancer; CRT: chemoradiotherapy; IMAC: intramuscular adipose tissue content; mIMAC: modified IMAC; PD: psoas density; CC: colon cancer; RC: rectal cancer; SFD: subcutaneous fat distance; SFI: subcutaneous fat index; SMD: skeletal muscle density, SMI: skeletal muscle index; SSI: surgical site infection; TSM: total skeletal muscle; TBF: total body fat; TPA: total psoas area; TAMA: total abdominal muscle area; PMI: psoas muscle index; SMA: skeletal muscle area; TFI: total fat index; TPI: total psoas muscle index; VAT: visceral adipose tissue; VFI: visceral fat index.


Furthermore, when assessing the evidence, we may need to take the biological differences between colon cancer and rectal cancer into consideration. Colon cancer and rectal cancer have traditionally been recognized as clinically equivalent types of cancer for treatment purposes, and almost all of the evidence on myopenia and myosteatosis mentioned in this review comes from patients with either colon cancer or rectal cancer, as described in Table 1. However, a proposed consensus molecular subtypes revealed distinguished features of primary CRC in a location-dependent pattern [119]. Furthermore, additional analysis of clinical trial data has revealed that the effectiveness of molecular targeted agents, such as anti-epidermal growth factor receptor and anti-vascular endothelial growth factor, depends on whether CRC is right-sided or left-sided [120,121]. Considering the evidence for embryological differences between the colon and rectum, it has been gradually recognized that colon cancer and rectal cancer require different treatment strategies. Although several studies have assessed the clinical impact of myopenia and myosteatosis in colon cancer and rectal cancer, there is still not enough evidence in this regard, and further studies are needed to elucidate these issues.

Finally, we need to understand the pathogenesis of the intimate correlations between sarcopenia and oncological outcome, effect of chemoradiotherapy, and perioperative infectious complications in patients with CRC. Several cofounding factors in the cancer-harboring host, including diabetes, cardiovascular disease, chronic obstructive pulmonary disease, aging, frailty, injury, obesity, colitis, and other chronic inflammatory disease, could also be drivers of sarcopenia and could in themselves increase susceptibility to several comorbidities during perioperative and postoperative treatment of CRC. Perioperative infectious complications and adverse events stemming from chemotherapy or radiotherapy could also have a negative effect on the survival outcome in patients with CRC. These cofounding factors in the host could reduce muscularity further as a consequence of the adverse effects of chemoradiotherapy and the progression of cancer and contribute to a poor survival outcome in these patients. In contrast, several factors that can cause sarcopenia also have a pivotal role in the progression of cancer, and cancer-induced inflammation is a well-known factor in the pathogenesis of both oncogenesis and sarcopenia in patients with malignancy. Sarcopenia-induced factors related to the cancer, host, cancer–host interaction, and sarcopenia itself might be bidirectionally correlated, thereby exacerbating low muscularity and increasing the likelihood of a poor survival outcome in patients with CRC.


## 11. Conclusions

The evidence presented here indicates that low muscularity is associated with various negative outcomes in patients with CRC. However, there are still no definitive criteria for the diagnosis of myopenia or myosteatosis using CT imaging in patients with malignancy, including CRC. Nevertheless, there is a significant body of literature on exercise and dietary strategies to improve sarcopenia and cachexia with the aim of improving functional quality of life [92,122,123,124,125,126,127]. One of the potential benefits of the CT assessment of low muscularity is that it would allow for the objective quantification of muscle loss at sequential time points during the treatment of CRC. Multiple CT assessments using optimal cut-off values could allow timely nutritional intervention and exercise therapy to improve outcomes in patients with CRC.

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
