# Peer review of "Clinical Relevance of Myopenia and Myosteatosis in Colorectal Cancer"

_jcm, 2022, doi:10.3390/jcm11092617_

Round 1
Reviewer 1 Report
I enjoyed reading this review, the text is simple, but it brings together the main studies associating muscle mass and clinical outcomes in patients with CRC.
I suggest some topic insertions to improve the quality of the review.
1. Insert a topic on the pathophysiology and molecular mechanisms of muscle mass loss in cancer.
2. Discuss the different ways of evaluating low muscularity through CT among the papers. The use of different regions (T12, L1, L3, L4), muscle groups and ways to establish the cut-off points..
3. Include a table showing those differences in methodology to define low muscularity/sarcopenia between articles.
4. Differentiate the terms sarcopenia and low muscularity throughout the text.
5. Discuss the differences between colon cancer and rectal cancer. Some articles assess colon cancer only, others only rectal cancer, or both. Discuss how this can impact the clinical outcomes evaluated.
6. Update reference 1.
Author Response
Reviewer #1: I enjoyed reading this review, the text is simple, but it brings together the main studies associating muscle mass and clinical outcomes in patients with CRC. I suggest some topic insertions to improve the quality of the review.
- Insert a topic on the pathophysiology and molecular mechanisms of muscle mass loss in cancer.
Response: We real appreciate for both reviewer’s positive comments. We completely agree with reviewer’s suggestion, and insert recent topics on the pathophysiology and molecular mechanisms of muscle mass loss in cancer (Page5, Line4 – Page6, Line11).
- Discuss the different ways of evaluating low muscularity through CT among the papers. The use of different regions (T12, L1, L3, L4), muscle groups and ways to establish the cut-off points.
Response: In agreement with this comment (#1-2) with another comment (#1-3), we included additional information about the different regions and muscle group using CT image in revised Table 1. Furthermore, we discussed these points as one of current limitations in revised manuscript (Page17, Line1–14, revised Table 1).
- Include a table showing those differences in methodology to define low muscularity/sarcopenia between articles.
Response: According to reviewer’s comment, we included the differences in methodology to define low muscularity/ sarcopenia in revised Table 1.
- Differentiate the terms sarcopenia and low muscularity throughout the text.
Response: In agreement with this reviewer’s comment, we rechecked and edit to differentiate the terms sarcopenia and low muscularity throughout the text (Page1, Line2-3; Page4, Line20-22; Page5, Line1; Page6, Line22; Page7, Line3; Page7, Line5; Page7, Line8; Page8, Line2-3; Page8, Line6-7; Page9, Line3; Page9, Line6; Page10, Line2-4; Page10, Line6; Page10, Line11; Page10, Line14; Page10, Line17; Page10, Line19; Page11, Line4; Page12, Line1; Page12, Line6; Page12, Line8; Page12, Line11; Page12, Line19; Page13, Line8; Page13, Line10; Page13, Line13; Page13, Line19; Page14, Line1-3; Page14, Line5; Page14, Line8; Page16, Line19).
- Discuss the differences between colon cancer and rectal cancer. Some articles assess colon cancer only, others only rectal cancer, or both. Discuss how this can impact the clinical outcomes evaluated.
Response: We absolutely agree with reviewer’s precious suggestion, and we added discussion about the differences between colon and rectal cancer to clarify how this can impact the clinical outcome in each type of cancer (Page17, Line15 – Page18, Line7).
- Update reference 1.
Response: I sincerely apologize for lack of update about reference 1. I updated about reference 1 from global cancer statistics 2018 to global cancer statistics 2020 in revised manuscript (Page22, Line2-5).

Reviewer 2 Report
The authors seek to review the literature examining the impact of sarcopenia on colorectal cancer survivorship. This is an important area of discussion; however, the are significant omissions that need to be addressed. Additionally, the overall goal and scope of the review is unclear
First, it is not clear what your outline is. The entire review is under 1 major heading title "1. Introduction". Everything thereafter is a "1.X" which seems pointless then to have any designation at all. It seems there are natural groupings that could be made, but are not.
Next, sarcopenia as a pre-diagnosis/pre-treatment phenomena is impactful and relevant to understand; however, there is no discussion on the overlap of sarcopenia and cachexia, the loss of mass during cancer and cancer treatment. Low pretreatment body weight will increase your susceptibility to several comorbidities, increased bodyweight loss throughout cancer's progression would exacerbate these outcomes.
Next, There is no speculation or discussion of how (mechanistically) sarcopenia would impact survival (or other) outcomes, but rather the authors simply sate that those who are sarcopenic are worse off. There is no nuance to any of the other contributing factors to what drives an individual to become sarcopenic (independent of cancer) which may impact colon cancer prognosis (i.e. diabetes, CVD, Frailty, Injury, Obesity, Colitis, etc.).
Last, the conclusions are lacking. Based on the argument presented it is unclear how coming up with yet again another definition of sarcopenia would improve treatment outcomes or the patient condition. Based on the argument presenting it is convincing that sarcopenia (as currently defined) is negatively associated with CRC patient outcomes. Simply recommending more definitions would not provide insight or clarity into the condition, especially when no therapies or interventions are postulated. There is significant body literature looking at exercise or different dietary strategies to improve sarcopenia or cachexia aimed at improving functional quality of life.
Author Response
Reviewer #2: The authors seek to review the literature examining the impact of sarcopenia on colorectal cancer survivorship. This is an important area of discussion; however, the are significant omissions that need to be addressed. Additionally, the overall goal and scope of the review is unclear.
Response: I really appreciate for this reviewer’s precious suggestion. I sincerely agree with these comments, and edited the revised manuscript according to reviewer’s comments described below.
First, it is not clear what your outline is. The entire review is under 1 major heading title "1. Introduction". Everything thereafter is a "1.X" which seems pointless then to have any designation at all. It seems there are natural groupings that could be made, but are not.
Response: We have to apologize for the complicated outline to be difficult to understand for readers. In agreement with this comment, we changed the all of outlines and exclude number of heading title to be seems natural grouping in revised manuscript.
Next, sarcopenia as a pre-diagnosis/pre-treatment phenomena is impactful and relevant to understand; however, there is no discussion on the overlap of sarcopenia and cachexia, the loss of mass during cancer and cancer treatment. Low pretreatment body weight will increase your susceptibility to several comorbidities, increased bodyweight loss throughout cancer's progression would exacerbate these outcomes.
Response: We absolutely agree with reviewer’s precious suggestion. In agreement with these comments, we added the discussion including overlap of sarcopenia and cachexia, the loss of mass during cancer treatment, and negative impact of sarcopenia about susceptibility to several comorbidities in patients with malignancies (Page14, Line17 – Page15, Line3, Page18, Line8 - Page19, Line3).
Next, There is no speculation or discussion of how (mechanistically) sarcopenia would impact survival (or other) outcomes, but rather the authors simply sate that those who are sarcopenic are worse off. There is no nuance to any of the other contributing factors to what drives an individual to become sarcopenic (independent of cancer) which may impact colon cancer prognosis (i.e. diabetes, CVD, Frailty, Injury, Obesity, Colitis, etc.).
Response: We absolutely agree with reviewer’s suggestion. It still remains unclear how sarcopenia would mechanistically impact survival and other outcomes in CRC patients. However, complicated complex between direct mechanism and indirect cofounding factors might be influences these negative outcomes in CRC patients. In agreement with reviewer’s comment, we discussed these points in discussion section of revised manuscript (Page18, Line8- Page19, Line3).
Last, the conclusions are lacking. Based on the argument presented it is unclear how coming up with yet again another definition of sarcopenia would improve treatment outcomes or the patient condition. Based on the argument presenting it is convincing that sarcopenia (as currently defined) is negatively associated with CRC patient outcomes. Simply recommending more definitions would not provide insight or clarity into the condition, especially when no therapies or interventions are postulated. There is significant body literature looking at exercise or different dietary strategies to improve sarcopenia or cachexia aimed at improving functional quality of life.
Response: We agree with reviewer’s suggestion. According to reviewer’s precious comment, we added the conclusion to discuss these points in revised manuscript (Page19, Line5–15).

Round 2
Reviewer 2 Report
All of my comments have been adequately addressed.